# Functional Analysis of GSTK1 in Peroxisomal Redox Homeostasis in HEK-293 Cells

**DOI:** 10.3390/antiox12061236

**Published:** 2023-06-07

**Authors:** Cláudio F. Costa, Celien Lismont, Serhii Chornyi, Hongli Li, Mohamed A. F. Hussein, Hans R. Waterham, Marc Fransen

**Affiliations:** 1Laboratory of Peroxisome Biology and Intracellular Communication, Department of Cellular and Molecular Medicine, Katholieke Universiteit Leuven, 3000 Leuven, Belgium; claudiofcosta@kuleuven.be (C.F.C.); celien.lismont@kuleuven.be (C.L.); hongli.li@kuleuven.be (H.L.); mohamed.hussein@kuleuven.be (M.A.F.H.); 2Laboratory Genetic Metabolic Diseases, Department of Clinical Chemistry, Amsterdam University Medical Center, University of Amsterdam, 1105 AZ Amsterdam, The Netherlands; s.chornyi@amsterdamumc.nl (S.C.); h.r.waterham@amsterdamumc.nl (H.R.W.); 3Department of Biochemistry, Faculty of Pharmacy, Assiut University, 71515 Asyut, Egypt

**Keywords:** peroxisome, glutathione, glutathione S-transferase, GSTK1, oxidative insult, redox state recovery, glutaredoxin

## Abstract

Peroxisomes serve as important centers for cellular redox metabolism and communication. However, fundamental gaps remain in our understanding of how the peroxisomal redox equilibrium is maintained. In particular, very little is known about the function of the nonenzymatic antioxidant glutathione in the peroxisome interior and how the glutathione antioxidant system balances with peroxisomal protein thiols. So far, only one human peroxisomal glutathione-consuming enzyme has been identified: glutathione S-transferase 1 kappa (GSTK1). To study the role of this enzyme in peroxisomal glutathione regulation and function, a GSTK1-deficient HEK-293 cell line was generated and fluorescent redox sensors were used to monitor the intraperoxisomal GSSG/GSH and NAD^+^/NADH redox couples and NADPH levels. We provide evidence that ablation of GSTK1 does not change the basal intraperoxisomal redox state but significantly extends the recovery period of the peroxisomal glutathione redox sensor po-roGFP2 upon treatment of the cells with thiol-specific oxidants. Given that this delay (i) can be rescued by reintroduction of GSTK1, but not its S16A active site mutant, and (ii) is not observed with a glutaredoxin-tagged version of po-roGFP2, our findings demonstrate that GSTK1 contains GSH-dependent disulfide bond oxidoreductase activity.

## 1. Introduction

Complex living systems can only survive within a very narrow range of physicochemical conditions [1]. However, they are constantly exposed to a myriad of stimuli that disrupt this equilibrium [2]. To not endanger life, organisms have developed and rely on intricate signaling networks to return to the homeostatic state [1]. An inefficiency of these networks can lead to illness and ultimately death, thereby highlighting their importance for proper cell functioning [2]. From a cellular perspective, a relevant example of a healthy physiological state is redox homeostasis, which refers to the dynamic equilibrium between molecules that regulate the electrophilic and nucleophilic tone [3].

An important molecule for redox homeostasis is the tripeptide γ-glutamyl-cysteinyl-glycine, commonly known as glutathione, whose thiol group supplies reducing power for numerous cellular redox reactions [4]. This metabolite is part of an intricate thiol-dependent antioxidant defense mechanism, known as the glutathione system, which depends on NADPH and a group of enzymes that regulate the amount of reduced and oxidized glutathione [5]. Glutathione is an essential metabolite that serves a plethora of biological functions [6]. Specifically, its reduced form (GSH) can (i) provide reducing power to scavenge oxidants such as hydrogen peroxide (H_2_O_2_) and lipid peroxides (LOOH), thereby being converted to a disulfide-bonded glutathione dimer (GSSG) [7], (ii) modify cysteine thiol groups via S-glutathionylation, a reversible posttranslational modification that can modulate the activity and/or the function of a protein and protect protein thiols from irreversible oxidative damage [8], (iii) detoxify electrophilic xenobiotics and endobiotic noxious compounds through a glutathione S-transferase (GST)-catalyzed reaction, thereby rendering them more water soluble prior to elimination [6,9], (iv) provide reducing power to break protein disulfide bridges, thereby also being oxidized to GSSG [5], and (v) act as a cofactor for GSTs that display ketosteroid isomerase activity [10].

Peroxisomes are organelles that play a pivotal role in cellular lipid metabolism [11]. Given that these organelles contain multiple H_2_O_2_-producing oxidases as well as catalase (CAT), a major H_2_O_2_-decomposing enzyme, their activity also contributes to cellular redox balancing and regulation [12]. However, very little is currently known about the presence and metabolism of glutathione in mammalian peroxisomes. For example, although GSH has been found in isolated human fibroblast peroxisomes [13], it is questionable whether peroxisomes contain glutathione peroxidase (GPX) activity [13,14], and no peroxisomal GPXs have been identified at the molecular level so far. Similarly, despite the fact that no peroxisomal glutaredoxins (GRXs) have been described to date, it has been demonstrated that roGFP2, a fluorescent GSSG/GSH sensor whose principle of action depends on the formation of a disulfide crosslink between two cysteines near the chromophore, can be reverted back to its reduced form within the peroxisomal matrix, implying GRX-like activity in loco [15]. Furthermore, the mechanism by which GSH can be recycled from GSSG inside peroxisomes remains a mystery. In other subcellular compartments, such as the cytosol and mitochondria, this process is catalyzed by a glutathione reductase, using NADPH as reducing power [16], but no such enzyme has been described in peroxisomes [12,13]. Currently, glutathione S-transferase kappa 1 (GSTK1) is the only enzyme of glutathione metabolism that has been identified in the mammalian peroxisomal proteome [17,18].

GSTK1, also sometimes referred to as disulfide-bond A oxidoreductase-like protein (DSBA-L), is a dimeric member of the glutathione S-transferase protein family that diverges from almost all other human GSTs in terms of sequence similarity and protein structure [19,20,21]. Originally described in the mitochondria of rodents [22,23,24], the presence of a peroxisomal targeting signal 1 (PTS1) enabled its subsequent identification as a peroxisomal protein [17,25]. More recently, GSTK1 was also proposed to be located in the endoplasmic reticulum (ER) [26]. While purified GSTK1 has been demonstrated to exhibit (i) GSH-conjugating activity towards halogenated aromatics, such as 1-chloro-2,4-dinitrobenzene and 4-nitrobenzylchloride [17,23,27], and (ii) peroxidase activity towards cumene hydroperoxide, tert-butyl hydroperoxide, and 15-S-hydroperocy-5,8,11,13-eicosatetraenoic acid [17], other studies in the cancer and fertility fields have suggested that the enzyme may also be important to conjugate GSH to cisplatin and arsenic-based compounds, respectively [28,29,30]. In addition, in cellulo studies have demonstrated that GSTK1 provides antioxidant protection against lipid peroxides [31]. In the ER, GSTK1 was proposed to play a role in the multimerization and stabilization of adiponectin [26,32,33], although this is still under debate [34,35]. From a macro perspective, a lack of GSTK1 has been linked to hypertrophic cardiomyopathy in zebrafish [36] and diabetic renal tubular injury [37], subclinical glomerular disease [34], and obesity-induced inflammation and insulin resistance [38] in mice. In patients, a decrease in GSTK1 has been associated with accelerated chronic kidney disease progression [37] and aging-related declines in respiratory performance [39]. On the other hand, fat-specific GSTK1 overexpression shields mice from diet-induced obesity, insulin resistance, and hepatic steatosis [32], and increased GSTK1 levels have been linked to colon cancer progression [40] and improved survival in patients with luminal B breast cancer [41].

This study focuses on the role of GSTK1 in peroxisomal glutathione homeostasis. Given that (i) GSTK1 makes use of GSH as an obligate cofactor or cosubstrate [24,42], (ii) GSH and NAD(P)H levels often correlate [43,44,45], (iii) GSH is a crucial substrate for many enzymes involved in H_2_O_2_ detoxification [7], and (iv) depending on the cell type, its physiological state, and the microenvironment, peroxisomes can be a significant source or sink of H_2_O_2_ [12]. We examined how GSTK1 inactivation in human embryonic kidney 293 (HEK-293) cells impacted these redox metabolites in peroxisomes and the cytosol under both basal and/or oxidative stress conditions. Since the loss of GSTK1 impairs the intraperoxisomal redox recovery of roGFP2 after an oxidative insult, and this trait can be reversed by expressing po-GRX1-roGFP2, our findings demonstrate that the peroxisome-associated pool of GSTK1 exhibits glutathione-disulfide oxidoreductase activity in cellulo.

## 2. Materials and Methods

### 2.1. Plasmids and DNA Manipulations

The plasmids encoding peroxisomal (po-) roGFP2 [15], cytosolic (c-) roGFP2 [15], po-roGFP2-Orp1 [46], c-roGFP2-Orp1 [46], po-GRX1-roGFP2 [47], po-mKeima [48], po-HsGSTK1-roGFP2 [31], po-HsGSTK1_S16A_-roGFP2 [31], and po-mCherry [49] have been described elsewhere. The plasmids coding for HsGSTK1 or HsGSTK1_S16A_ were generously supplied by Dr. Markus Kunze (Medical University of Vienna, Vienna, Austria). The pSpCas9(BB)-2A-GFP [50] plasmid was kindly provided by Dr. Feng Zhang (Massachusetts Institute of Technology, USA), and the plasmids coding for c-SoNar [51], c-cpYFP [51], c-iNAP1 [52], and c-iNAPc [52] were provided by Dr. Yi Yang (East China University of Science and Technology, Shanghai, China).

The plasmids encoding po-SoNar, po-iNAP1, or po-iNAPc were generated by amplifying the corresponding coding sequences via polymerase chain reaction (PCR): oligonucleotides for po-SoNar, 5′-cgag**ggatcc**accatgggcaaccggaagtggggcctgtgc-3′ and 5′-ggggg**gcggccgc**tcacagtttggacttgcccatcatctcctcccgccac-3′; and for po-iNAP1 and po-iNAPc, 5′-cgag**ggatcc**accatggatccgatgaaccgg-3′ and 5′-cgaggg**gcggccgc**tcacagtttggacttgcccatcatctcctccc-3′ (the restriction sites are indicated in bold; the nucleotides encoding the PTS1 are underlined), and ligating the *Bam*HI/*Not*I-digested PCR products into similarly-digested pEGFP-N1 (Clontech, 6085-1). The plasmids encoding po-HsGSTK1-mCherry or po-HsGSTK1_S16A_-mCherry were generated by amplifying the po-mCherry coding sequence via PCR (oligonucleotides: 5′-cgagag**aagctt**accatggtctctaagggcga-3′ and 5′- ggcggg**gcggccgc**ttacagtttagatttgtacagttcatcc-3′) and ligating the *Hin*dIII/*Not*I-digested PCR product into the backbones of the similarly digested po-HsGSTK1-roGFP2- and po-HsGSTK1_S16A_-roGFP2-encoding plasmids, respectively. To construct the GSTK1 guide RNA-encoding pSpCas9(BB)-2A-GFP plasmids, two oligonucleotides (5′-caccgtgcggcccagcctcataaca-3′ and 5′-aaactgttatgaggctgggccgcac-3′; the guide sequences are underlined) were annealed and subsequently cloned into the *Bpi*I-digested pSpCas9(BB)-2A-GFP plasmid.

The *TOP10F’ E. coli* strain (Thermo Fisher Scientific, Dilbeek, Belgium; C3030-06) was used as a cloning and plasmid amplification host. Oligonucleotides (Integrated DNA Technologies, Leuven, Belgium), restriction enzymes (Takara Bio Europe SAS, Saint-Germain-en-Laye, France), and Platinum *Pfx* DNA polymerase (Invitrogen, Merelbeke, Belgium; 11708039) were commercially obtained, and the correctness of the plasmid inserts was verified by Sanger sequencing (LGC Genomics, Berlin, Germany).

### 2.2. Cell Culture, Transfections, Treatments, and Subcellular Fractionation

The DD-DAO Flp-In T-REx 293 cell line routinely used in this research (here referred to as HEK-293) has been described elsewhere [46]. Cells were routinely cultured in regular minimum essential medium Eagle α (rMEMα; Minimum Essential Medium (Thermo Fisher Scientific, M4655) supplemented with 10% (*v*/*v*) fetal bovine serum (FBS; Avantor, Leuven, Belgium, Biowest S181B), 2 mM Ultraglutamine-1 (Lonza, Verviers, Belgium; BE17-605E/U1), and 0.2% (*v*/*v*) Mycozap (Lonza, VZA-2012)) at 37 °C in a humidified 5% CO_2_ incubator. HEK-293 cells were electroporated (1150 V, 20-ms pulse width, 2 pulses) with the Neon Transfection System (Thermo Fischer Scientific) in combination with a homemade sucrose-based RF1 buffer [53].

To selectively generate H_2_O_2_ inside peroxisomes through expression and activation of a destabilization domain (DD) tagged version of D-amino acid oxidase (DAO), the HEK-293 cells were processed as described elsewhere [46]. For other cell treatments, 4,4′-dithiodipyridine (aldrithiol-4, AT-4; Thermo Fisher Scientific, 10607332; in ethanol (Thermo Fisher Scientific, 12498740)), 1,1′-azobis(N,N-dimethylformamide) (diamide; Fluorochem, Amsterdam, The Netherlands; 046719; in water), H_2_O_2_ (Chemlab, Zedelgem, Belgium; CL00.2306.1000; in water), Triton X-100 (Sigma-Aldrich, St. Louis, MO, USA; T9284; in water), or Dulbecco’s phosphate-buffered saline (DPBS) containing 3-amino-1,2,4-triazole (3-AT; Thermo Fisher Scientific, 264571000; in water), D-alanine (D-Ala; Carl Roth, Karlsruhe, Germany; 7866.1), or L-alanine (L-Ala; Fluka Chemicals, Geel, Belgium; 05129) were added at the concentrations and time intervals shown in the figure legends.

Subcellular fractionations of HEK-293 cells were performed as described [54]. Based on this study, the gradient fractions with a density ranging between 1.20 and 1.08 g/mL (fractions 12 to 20) were processed for immunoblotting.

### 2.3. Generation of Gene Knockout Cells by CRISPR-Cas9 Genome Editing

The CRISPR-Cas9 genome editing technology (as described by [55]) was used to introduce a functional disruption of the *GSTK1* gene in DD-DAO Flp-In T-REx 293 cells [46]. To this end, the cells were transfected with the plasmid, encoding the guide RNA and clonal cells with homozygous (clone 2, cl 2; c.133insA) or compound heterozygous (clone 1, cl 1; c.133insA/c.133delA) out-of-frame variants were selected for downstream analysis (NCBI reference sequence: NM_015917.3).

### 2.4. Antibodies and Immunoblotting

The following primary antibodies were used for immunoblotting: rabbit anti-3-ketoacyl-CoA thiolase (ACAA1) (Atlas Antibodies, Stockholm, Sweden; HPA007244, 1:1000), anti-CAT (Calbiochem, Darmstadt, Germany; 219010, 1:1000), anti-cytochrome c oxidase subunit 4 (COX IV) (Cell Signaling Technology, Danvers, MA, USA; 4850, 1:1000), anti-enhanced green fluorescent protein (EGFP) (1:1000) [56], anti-peroxisomal multifunctional enzyme type 2 (HSD17B4) (Fisher Scientific, Merelbeke, Belgium; 15116-1-AP, 1:1000), anti-NADH-ubiquinone oxidoreductase chain 6 (ND6) (Santa Cruz Biotechnology, Heidelberg, Germany; sc-20667, 1:1000), anti-peroxin 5 (PEX5) (Sigma-Aldrich; HPA039259, 1:1000), anti-peroxin 14 (PEX14) (1:1000) [57], anti-peroxiredoxin 5 (PRDX5) (1:500) [58], anti-sequestosome 1 (SQSTM1) (Abcam, Cambridge, UK; ab109012, 1:20,000), and antitranslocase of outer mitochondrial membrane 22 (TOMM22) (Sigma-Aldrich; HPA003037, 1:1000); mouse anti-β-actin (ACTB) (Sigma-Aldrich; A5316, 1:1000), and anti-GSTK1 (Santa Cruz Biotechnology; sc-515580, 1:1000); goat anti-calreticulin (CALR) (Everest Biotech, Oxfordshire, UK; EB12387, 1:1500); and affinity isolated anti-goat (Sigma-Aldrich; A7650, 1:10,000), anti-rabbit (Sigma-Aldrich; A3687, 1:5000) and anti-mouse (Sigma-Aldrich; A2429, 1:10,000) antibodies conjugated to alkaline phosphatase.

Proteins samples were precipitated by treating DPBS-resuspended cell suspensions with 6% (*w*/*v*) trichloroacetic acid and 0.0125% (*w*/*v*) deoxycholate, followed by acetone-washing. Thereafter, the protein samples were processed for reducing or nonreducing SDS-PAGE and immunoblotting as described elsewhere [46]. Immunoblot signal intensities were quantified using ImageJ software [59].

### 2.5. Fluorescence Microscopy

Fluorescence microscopy was carried out as described previously [15,60] using the following filter cubes: F400 (excitation: 390–410 nm; dichroic mirror: 505 nm; emission: 510–550 nm), F425 (excitation: 422–432 nm; dichroic mirror: 600 nm; emission: 610 long pass), F480 (excitation: 470–495 nm; dichroic mirror: 505 nm; emission: 510–550 nm), and F565 (excitation: 545–580 nm; dichroic mirror: 600 nm; emission: 610 nm long pass). The cells were seeded and imaged in FluoroDish cell culture dishes (World Precision Instruments, Hertfordshire, UK; FD-35) precoated with 25 μg/mL polyethyleneimine (MP Biomedicals, 195444) [46], and cellSens Dimension software (version 2.1) (Olympus Belgium) was used for image acquisition and analysis.

### 2.6. Assessment of Mitochondrial Oxidative Phosphorylation

Cells were first cryopreserved in FBS supplemented with 10% dimethyl sulfoxide (DMSO; MP Biomedicals, Solon, OH, USA; 196055) [61]. For the assay, the cryopreserved cells were resuspended in MiR05 medium (Oroboros, Innsbruck, Austria) and mitochondrial respiratory parameters were measured using an Oroboros O2k high-resolution respirometer. The cells were first permeabilized with 5 µg/mL digitonin (Sigma-Aldrich, D5628) and then subjected to a shortened version of the SUIT-001 reference protocol [62]. Briefly, to the O2k chamber were added, subsequently and in this order, (i) control or ΔGSTK1 HEK-293 cells, (ii) 5 µg/mL digitonin (Sigma-Aldrich, D5628), (iii) 5 mM pyruvate (Sigma-Aldrich, P2256) and 2 mM malate (Sigma-Aldrich, M1000), 2.5 mM ADP (Calbiochem, 117105-1GM), (v) 10 μM cytochrome c (Sigma-Aldrich, C7752), (vi) 0.5 μM carbonyl cyanide m-chlorophenyl hydrazone (CCCP; Sigma-Aldrich, C2759), (vii) 10 mM glutamate (Sigma-Aldrich, G1626), (viii) 50 mM succinate (Sigma-Aldrich, S2378), (ix) 0.5 mM octanoylcarnitine (BioTrend, Cologne, Germany; B6371), and (x) 0.5 μM rotenone (Sigma-Aldrich, R8875). LEAK respiration was measured following the addition of pyruvate and malate. OXPHOS (routine) respiration was measured following the addition of ADP. The different electron-transfer (ET) capacities were measured after treating the cells with CCCP. Electron transfer was examined in the NADH-linked substrate (N-) pathway after the addition of glutamate, the N- and succinate (S-) pathway (NS-) after the addition of succinate, and the S-pathway after the addition of the complex I inhibitor rotenone. The addition of cytochrome c allowed to verify the integrity of the mitochondrial outer membrane.

### 2.7. Statistical Analysis

Statistical analysis was performed using GraphPad Prism (version 9.0.0 for Windows 64-bit, GraphPad Software, San Diego, CA, USA). The statistical tests used are specified in the figure legends. A *p*-value lower than 0.05 was considered statistically significant. Considering this is an exploratory study, no Bonferroni corrections were applied.

## 3. Results

### 3.1. GSTK1 Displays Dual Peroxisomal and Mitochondrial Localization in HEK-293 Cells

Given the contradictory findings on the subcellular location of GSTK1 [17,22,23,24,25,26], we first confirmed that at least a portion of GSTK1 cofractionates with peroxisomes in HEK-293 cells. As initial experiments indicated that our anti-GSTK1 antibody was not suitable for immunofluorescence microscopy, we subjected a post-nuclear supernatant from HEK-293 cells to Nycodenz density gradient centrifugation. Immunoblot analysis of the relevant organelle-containing fractions [54] demonstrated that GSTK1 clearly coenriches with the peroxisomal marker CAT and the mitochondrial protein TOMM22 (Figure 1), which is consistent with (i) predictions that the protein contains peroxisomal and mitochondrial targeting information within its mature protein sequence [25] and (ii) its previously described dual localization in mitochondria and peroxisomes in human hepatoblastoma cells [17]. In addition, despite observing a similar distribution pattern between GSTK1 and TOMM22 in the low-density fractions 17 to 20, our findings do not exclude the potential presence of GSTK1 within the ER.

### 3.2. Generation and Validation of the ΔGSTK1 HEK-293 Cell Lines

To study the potential role of GSTK1 in peroxisomal glutathione redox metabolism, we selectively disrupted the corresponding gene in HEK-293 cells (genetic background: DD-DAO Flp-In-T-REx 293 [46]) by using the CRISPR-Cas9 technology [55].

A heterozygous (c.133insA/c.133delA; cl 1) and a homozygous (c.133insA; cl 2) knockout clone were selected and their correctness was validated at the protein level by immunoblot analysis (Figure 2 and Appendix A). If considered meaningful, both clones were incorporated into the analyses to reduce the chance of drawing inaccurate conclusions solely based on a single clonal population of cells. Note that GSTK1 inactivation does not significantly impact the expression levels of any of the other examined peroxisome-related (e.g., the β-oxidation enzymes ACAA1 and HSD17B4, the antioxidant enzymes CAT and PRDX5, and the peroxins PEX5 and PEX14) (Figure 2 and Appendix A), mitochondrial (e.g., COX IV, ND6, and TOMM22) (Figure 3A and Appendix A), and ER (e.g., CALR) (Figure 3A and Appendix A) proteins. As GSTK1 is located both in peroxisomes and mitochondria, we also carried out high-resolution respirometry studies to assess mitochondrial function. Again, no significant differences could be observed between the control and ΔGSTK1 cells (Figure 3B).

### 3.3. GSTK1 Inactivation Does Not Affect the Basal Peroxisomal and Cytosolic Redox States

Given that (i) the overarching aim of this study was to examine the role of GSTK1 in peroxisomal glutathione redox homeostasis, (ii) organellar glutathione pools rely entirely on cytosolic glutathione import [63], and (iii) changes in (local) glutathione metabolism may have a direct impact on the levels of other redox metabolites [7,43,44,45], we first evaluated if GSTK1 inactivation affected the overall peroxisomal and cytosolic GSSG/GSH, NAD^+^/NADH, NADPH, and H_2_O_2_ levels under basal conditions. To monitor possible changes in these metabolites, we used compartment-specific forms of the ratiometric fluorescent redox sensors roGFP2 [64], SoNar [65], iNAP1 [65], and roGFP2-Orp1 [64], respectively. The typical subcellular distribution patterns for each of these reporter proteins are depicted in Appendix A. All four reporters can exist in different conformational states, each of which corresponds to a particular fluorescent excitation spectrum. In the case of SoNar, the excitation maximum is at 485 nm or around 420 nm, depending on whether it binds NAD^+^ or NADH [51]. Likewise, the excitation maximum of iNAP1 varies depending on whether it binds NADPH or not (±500 nm or ±420 nm, respectively) [52]. As a result, increased NAD^+^/NADH or NADPH levels lead to higher F480/F400 ratios for SoNar or iNAP1, respectively. Since iNAP1 and SoNar are pH sensitive, it is critical to conduct parallel measurements with sensors that react similarly to pH but not to NAD(H) or NADPH (e.g., iNAPc, cpYFP, or mKeima). In the case of roGFP2 and roGFP2-Orp1, oxidation results in the formation of an intramolecular disulfide bridge, which shifts the excitation maximum from 488 nm to 405 nm [66]. Hence, the greater the observed F400/F480 ratio, the higher the GSSG/GSH and H_2_O_2_ levels, respectively. No significant alterations in any of the examined redox metabolites could be found between control and ΔGSTK1 cells, neither in peroxisomes nor in the cytosol (Figure 4). Potential explanations for this lack of a discernible redox phenotype in ΔGSTK1 cells may include (i) a non-redox role of GSTK1, (ii) a functional redundancy between GSTK1 and other (unidentified) proteins, or (iii) GSTK1’s nonessential role in maintaining redox homeostasis under the conditions studied.

### 3.4. GSTK1 Aids in the Recovery of Peroxisomal RoGFP2 after Oxidative Insult Withdrawal

To investigate whether GSTK1 regulates antioxidant defenses in response to oxidative stress, we subjected po-roGFP2-expressing control and ΔGSTK1 cells to mild and severe transient oxidative insults. First, we selectively generated H_2_O_2_ inside peroxisomes through the expression and D-Ala-mediated activation of DD-DAO [46]. Unfortunately, despite the fact that considerable amounts of po-H_2_O_2_ were generated (Appendix A), this treatment was unable to sufficiently oxidize po-roGFP2 to reliably quantify potential differences in recovery of the reduced fraction of po-roGFP2 (Appendix A). Within the context of these experiments, it is important to note that (i) under basal conditions, po-roGFP2-Orp1 is almost entirely oxidized in HEK-293 cells, rendering this reporter unsuitable for monitoring changes in po-H_2_O_2_ production [46] and (ii) po-H_2_O_2_ can rapidly permeate across the peroxisomal membrane, thereby allowing us to monitor peroxisomal H_2_O_2_ production with c-roGFP2-Orp1 [54]. Second, we exposed the cells to exogenous H_2_O_2_, but once again, we failed to observe a significant rise in the oxidation state of po-roGFP2 (Appendix A), a finding in line with our previous observations that peroxisomes very well resist oxidative stress generated outside the organelle [15].

Given the apparent difficulty in producing insults that po-roGFP2 can detect, we transiently exposed the cells to AT-4 and diamide, two oxidants whose mode of action involves a direct reaction with thiols [67,68] and that have previously been shown to completely oxidize roGFP2 in minutes [69]. Upon addition of AT-4, po-roGFP2 was rapidly oxidized in both control and ΔGSTK1 cells (Figure 5A). Intriguingly, after washout of the oxidant, analysis of the sensor’s redox state at different time intervals revealed that po-roGFP2 recovered from AT-4-induced oxidation in the ΔGSTK1 cells at a considerably slower rate than in the control cells (Figure 5A). Importantly, reintroducing GSTK1 into the ΔGSTK1 cells restored this deficit (Figure 5B,D), thereby demonstrating that the observed phenotype is indeed caused by a defect in GSTK1 function.

Unexpectedly, diamide administration had no discernible effect on the baseline oxidation of po-roGFP2, even not at a concentration of 5 mM (Appendix A). Given this surprising finding, we also investigated the effects of this oxidant on c-roGFP2 in both control and ΔGSTK1 cells. As previously reported [69], the oxidant rapidly oxidized c-roGFP2 (Figure 6B). Interestingly, no differences in recovery rates could be observed between the control and ΔGSTK1 cells (Figure 6B). This finding, which was also confirmed in cells transiently challenged with AT-4 (Figure 5C), demonstrates that the reduced recovery rate of roGFP2 in GSTK1 cells is a peroxisome-specific phenomenon. To potentially increase the intraperoxisomal diamide concentration, we coincubated the cells for 10 min with this oxidant in the presence of 0.001% (*w*/*v*) Triton X-100, a nonionic detergent that can permeabilize cell membranes. Under those conditions, po-roGFP2 responded similarly as observed in AT-4-treated control and ΔGSTK1 cells (Figure 6A). Combined, these findings provide evidence that GSTK1 can catalyze thiol-disulfide exchanges between roGFP2 and the glutathione redox pair in peroxisomes. Given that, under the conditions employed, the response of po-roGFP2 required concentrations in the millimolar range (Figure 6A and Appendix A); whereas c-roGFP2 already responded to 50 µM of AT-4 or diamide (Figure 5C and Figure 6B), the peroxisomal membrane apparently constitutes a permeability barrier for diamide and AT-4.

### 3.5. Po-GRX1-roGFP2′s Recovery Rates Are Comparable in AT-4-Insulted Control and ΔGSTK1 Cells

RoGFP-based probes have been demonstrated to specifically equilibrate with the glutathione redox potential through the action of (endogenous) GRXs, which catalyze the thiol-disulfide exchange between the GSH-GSSG and roGFP2 redox pairs [70]. Given that (i) until now, no peroxisomal GRXs have been identified [18], and (ii) roGFP2-derived probes do respond to a reduction in the peroxisomal redox state [15], a phenomenon impaired in ΔGSTK1 cells (Figure 5 and Figure 6), we examined the recovery behavior of po-GRX1-roGFP2 [47] in AT4-insulted ΔGSTK1 cells. These studies showed that the po-roGFP2 recovery deficits observed in ΔGSTK1 cells can be recovered by a protein with glutathione-disulfide oxidoreductase activity (Figure 7), indicating a potential functional overlap between the activities of both enzymes.

### 3.6. The GSTK1-Mediated Recovery of Po-RoGFP2 Depends on Its Active-Site Serine Residue

Previous studies have identified serine-16 (S16) as a critical residue for GSTK1 activity [19,31,33]. To evaluate the importance of this amino acid residue in the thiol-disulfide exchange between roGFP2 and the glutathione redox pair in peroxisomes, we complemented ΔGSTK1 cells with plasmids encoding wild-type (WT) or S16 to alanine (S16A)-mutated versions of the protein (C-terminally tagged or not with mCherry and a strong PTS1) and monitored the recovery rate of po-roGFP2 after brief exposure to AT-4. Given that the recovery rates of po-roGFP2 were significantly better in cells complemented with GSTK1_WT_ or po-GSTK1_WT_-mCherry as compared to GSTK1_S16A_ or po-GSTK1_S16A_-mCherry (Figure 8A,C), the recovery phenotype is clearly regulated by the GSTK1′s enzymatic activity. Note that mutating the active site serine to alanine had no effect on the expression levels (Figure 8B) and peroxisomal localization (Figure 8D) of the GSTK1 proteins examined.

### 3.7. Oxidative Insults Do Not Induce Intra- or Intermolecular Protein Disulfide Bonds in GSTK1

Given that (i) human GSTK1 has two cysteine residues (C27 and C176), and (ii) the latter cysteine has been reported to be a glutathionylation target in the mouse orthologue [71], we also investigated if GSTK1 can form intra- or intermolecular disulfide bonds in cells exposed to po-H_2_O_2_ or AT-4. However, neither internal nor external oxidative insults caused noticeable disulfide bond-induced conformational changes, as evaluated by a redox electrophoretic mobility shift assay (Figure 9). Note that SQSTM1, a protein reported to undergo disulfide-linked oligomerization in response to oxidative insults [48,72], was included as a positive control.

## 4. Discussion

Proteins inside the peroxisome lumen must be constantly safeguarded against the potentially damaging effects of H_2_O_2_ [73], and one potential protective mechanism may include the glutathione redox system [7]. However, although previous studies with genetically encoded redox sensors have demonstrated that the peroxisomal glutathione pool is maintained in a reduced state [15,74], virtually nothing is known about the role and metabolism of this low molecular weight antioxidant inside peroxisomes. In this study, we set out to examine the function of the peroxisomal pool of GSTK1, the only glutathione metabolism-related enzyme discovered in the mammalian peroxisomal proteome to date [17,18].

We first confirmed that peroxisomes in HEK-293 cells do indeed contain GSTK1 (Figure 1), an observation in line with the subcellular staining patterns visible in images generated in the context of the Human Protein Atlas project (https://www.proteinatlas.org/ENSG00000197448-GSTK1/subcellular; accessed on 26 April 2023).

Next, we examined how GSTK1 inactivation affected the expression levels of a specific set of peroxisomal and mitochondrial proteins (Figure 2, Figure 3A and Appendix A), mitochondrial respiration (Figure 3B), and peroxisomal and cytosolic redox levels, but no significant differences could be identified in cells cultured under normal conditions (Figure 4). However, this may not come as a surprise considering that GSTK1^−/−^ mice had normal amounts of reduced and total glutathione in their livers and kidneys [34]. As others have shown that loss of GSTK1 exacerbates oxidative stress and tubular apoptosis in the kidneys from streptozotocin-induced diabetic mice [37], we also investigated its potential role in peroxisomal oxidative stress responses. From these studies, it is clear that GSTK1 enhances the thiol-disulfide exchange between po-roGFP2 and the glutathione redox pair following oxidative insult withdrawal (Figure 5 and Figure 6).

As the po-roGFP2 recovery phenotype cannot be observed in ΔGSTK1 cells expressing po-GRX1-roGFP2 (Figure 7), our findings indicate that the peroxisomal pool of GSTK1 possesses glutaredoxin-like activity. Given that (i) GRXs catalyze disulfide bond reduction via a dithiol (CXXC-requiring) or monothiol (CXXS-dependent) mechanism [5,75,76], and (ii) the active center of GSTK1 (S_16_XXS_19_; Ser_16_) is lacking a cysteine [77], this may appear counterintuitive. However, here it is crucial to note that also other proteins with thiol-independent GRX activity have been identified. Examples include the bacterial disulfide-bond oxidoreductase YfcG and the yeast transcriptional regulator Ure2. YfcG exhibits very robust glutathione (GSH)-dependent disulfide-bond reductase activity toward the model substrate 2-hydroxyethyl disulfide [78] and Ure2 is a multifunctional protein with GRX activity toward small molecule disulfides (or GSH mixed disulfide bonds) and protein disulfides [79]. In addition, it has been proposed that these thiol-independent GRX-like activities are physiologically important under conditions of acute oxidative stress, where active site cysteines may be rapidly oxidized [79].

In summary, this study provides evidence that the peroxisomal pool of GSTK1 possesses GRX-like activity. This implies that, after the GST-Omega class [80], the GST-Kappa class becomes the second subfamily of human GSTs with such activity. It is noteworthy that the true endogenous substrates and mode of action of GSTK1 have yet to be identified. In the BioGRID database [81], only two peroxisomal proteins (alkyldihydroxyacetonephosphate synthase and CAT) have been reported to be physical interactors of GSTK1, a fact that may be explained by the transient nature of the GSTK1-protein substrate interactions or the absence of suitable interaction conditions. Another intriguing but unresolved question is why the recovery kinetics of po-roGFP2 and po-GRX1-roGFP2 are slower than that of c-roGFP2. However, given that little is known about how glutathione is transported across the peroxisomal membrane and how GSSG is regenerated to GSH within the peroxisome lumen, this is a challenging question. Nonetheless, given that (i) peroxisomes appear to be closed structures under in vivo conditions [82,83], and (ii) the transport of glutathione across the peroxisomal membrane requires a specific transporter in yeast [84], it is tempting to speculate that the delayed recovery phenotype may be due to a reduced replenishment of the peroxisomal GSH content following a strong oxidative insult.

## 5. Conclusions

In this study, we investigated the role of GSTK1 in peroxisomal redox metabolism in HEK-293 cells. We provide evidence that, while the protein appears to be unnecessary for peroxisomal redox homeostasis under basal conditions, the peroxisomal pool of GSTK1 possesses a glutaredoxin-like activity towards the glutathione redox sensor roGFP2. The discovery of this new biochemical function of GSTK1 opens up new avenues to advance our understanding of how changes in its expression contribute to aging and disease.

## Figures and Tables

**Figure 1 antioxidants-12-01236-f001:**
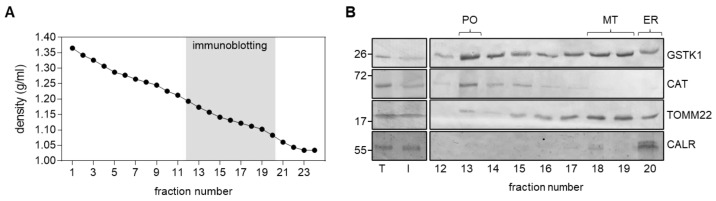
Subcellular fractionation of HEK-293 cells. Total cell homogenates (T) from HEK-293 cells were fractionated by differential centrifugation to yield a 1500× *g* supernatant (Input, I) that was subsequently subjected to Nycodenz gradient centrifugation. (**A**) Densities of the different gradient fractions. (**B**) Equivalent volumes of the T and I fractions and equal volumes of each gradient fraction were processed for immunoblotting with antibodies directed against GSTK1, catalase (CAT; peroxisomes, PO), translocase of outer mitochondrial membrane 22 (TOMM22; mitochondria, MT), and calreticulin (CALR; endoplasmic reticulum, ER). The signals observed for I represent 8% of the amount present in the gradient fractions. The migration points of relevant molecular mass markers (expressed in kDa) are shown on the left.

**Figure 2 antioxidants-12-01236-f002:**
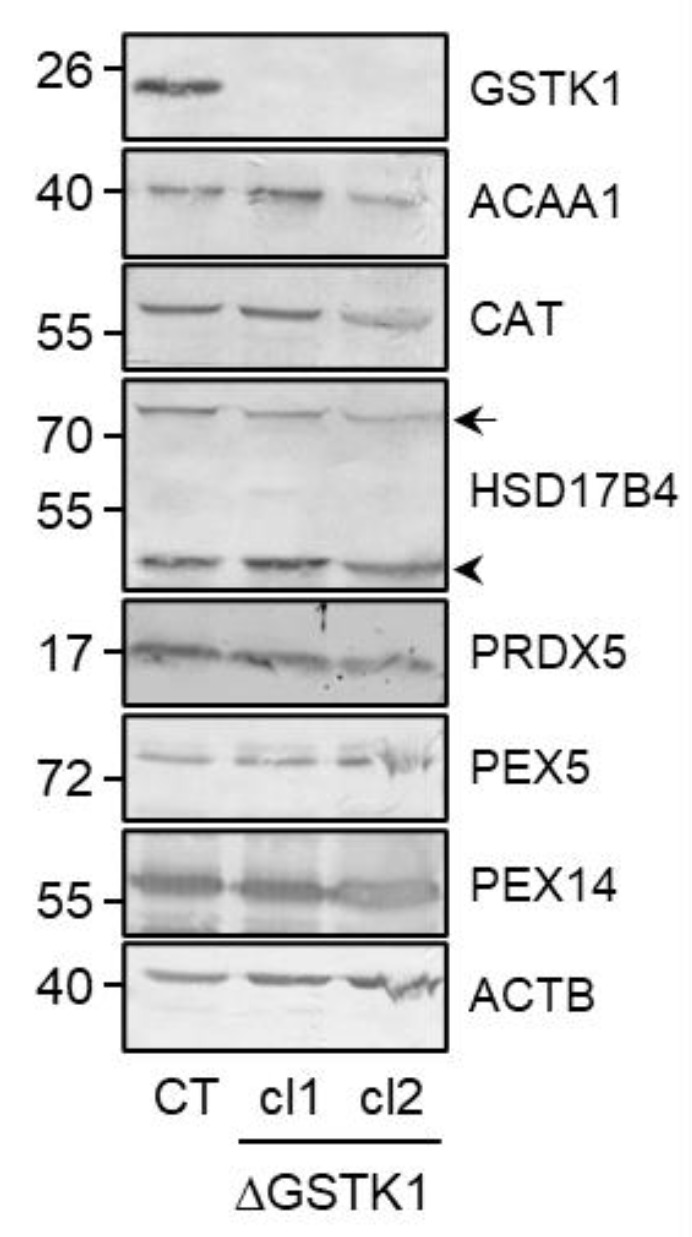
The absence of GSTK1 does not impact the expression levels of peroxisome-related proteins. Total cell lysates from control (CT), ∆GSTK1 (clone 1, cl1), or ∆GSTK1 (clone 2, cl2) cells, all containing equal amounts of protein, were processed for immunoblot analysis with antisera directed against the indicated proteins. Representative immunoblots are shown (immunoblot quantifications of three biological replicates are provided in Appendix A). The migration points of relevant molecular mass markers (expressed in kDa) are shown on the left. The arrow and arrowhead indicate the nonprocessed and processed forms of HSD17B4, respectively.

**Figure 3 antioxidants-12-01236-f003:**
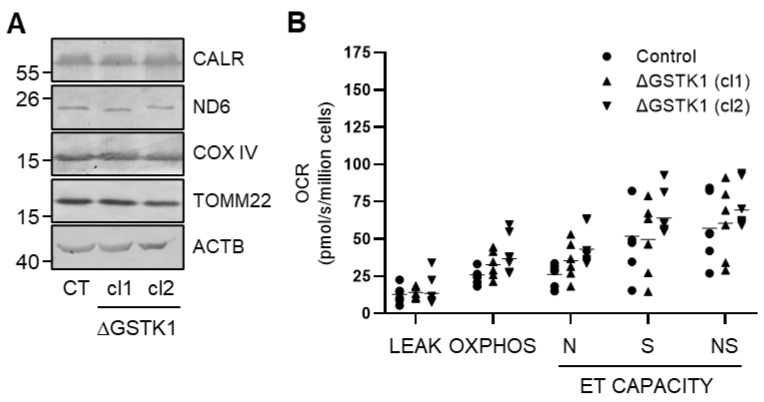
Loss of GSTK1 neither alters the expression levels of mitochondria-related proteins nor impacts mitochondrial oxidative phosphorylation. (**A**) Total cell lysates from control (CT), ∆GSTK1 (clone 1, cl1), or ∆GSTK1 (clone 2, cl2) cells, all containing equal amounts of protein, were processed for immunoblot analysis with antisera directed against the indicated proteins. Representative immunoblots are shown (immunoblot quantifications of three biological replicates are provided in Appendix A). The migration points of relevant molecular mass markers (expressed in kDa) are shown on the left. (**B**) The graph depicts the different oxygen consumption rate (OCR)-related parameters normalized to cell number. Each dot or triangle corresponds to an individual data point. The means are represented by horizontal lines. The data for each ΔGSTK1 cell line were statistically compared with those of the CT cell line using the ordinary two-way ANOVA test with Tukey’s multiple comparisons test, but no significant differences were found.

**Figure 4 antioxidants-12-01236-f004:**
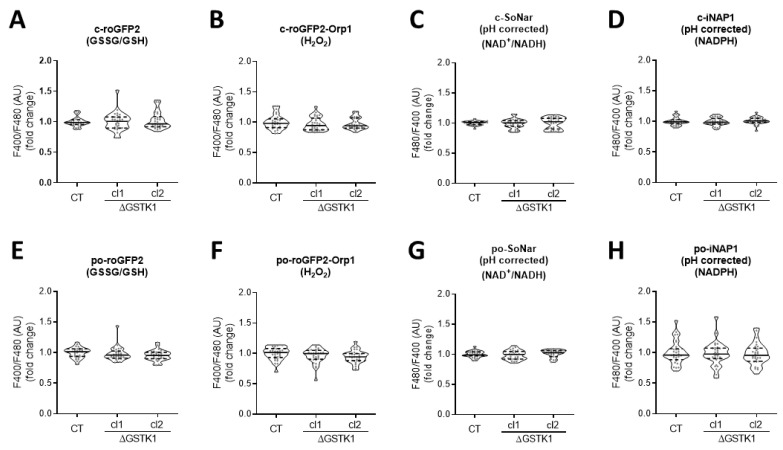
GSTK1 is not required for the maintenance of the basal peroxisomal and cytosolic redox states. CT or ∆GSTK1 (cl 1 and 2) HEK-293 cells were transfected with a plasmid encoding (**A**) c-roGFP2, (**B**) c-roGFP2-Orp1, (**C**) c-SoNar or c-cpYFP, (**D**) c-iNAP1 or c-iNAPc, (**E**) po-roGFP2, (**F**) po-roGFP2-Orp1, (**G**) po-SoNar or po-mKeima, or (**H**) po-iNAP1 or po-iNAPc and cultured in rMEMα. Two to three days later, the F400/F480 (for roGFP2 and roGFP2-Orp1) and pH-corrected F480/F400 response ratios (for SoNar and iNAP1) were measured, normalized to the average value of the control cells, and shown as violin plots. The data were derived from 30 individual images (represented as dots) from 3 independent experiments, each comprising 10 fields of view. The horizontal solid and dashed lines denote the median and the first and third quartiles, respectively. The data for each ΔGSTK1 cell line were statistically compared with those of the CT cell line using the Kruskal–Wallis test with Dunn’s multiple comparisons test, but no significant differences were found.

**Figure 5 antioxidants-12-01236-f005:**
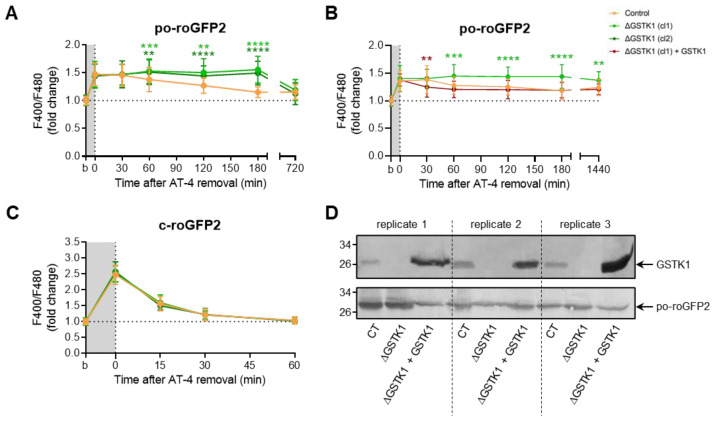
GSTK1 aids in the recovery of po- but not c-roGFP2 after aldrithiol-4 withdrawal. CT or ∆GSTK1 (cl 1 or 2) HEK-293 cells were transfected with a plasmid encoding po-roGFP2 or c-roGFP2, in combination or not with a plasmid encoding GSTK1, and cultured in rMEMα. (**A**–**C**) Two to three days later, the basal oxidation states (b) of po- and c-roGFP2 were measured and the cells were incubated with 50 µM (for c-roGFP2) or 5 mM (for po-roGFP2) aldrithiol-4 (AT-4) (grey background). After 10 min, the oxidant was removed and the response ratios of the sensors were monitored over time. The response values were normalized to the average value of the corresponding basal condition. The data points and vertical bars represent the mean and standard deviation of 30 to 60 individual measurements from 3 to 6 independent experiments, respectively. For each time point, the data obtained for the ∆GSTK1 conditions were statistically compared with those of the CT cells using the ordinary two-way ANOVA test with Tukey’s multiple comparisons test (**, *p* < 0.01; ***, *p* < 0.001; ****, *p* < 0.0001). (**D**) Total cell lysates from cells similar to the ones used in panel B were processed for immunoblot analysis with antisera directed against GSTK1 or EGFP. The migration points of relevant molecular mass markers (expressed in kDa) are shown on the left.

**Figure 6 antioxidants-12-01236-f006:**
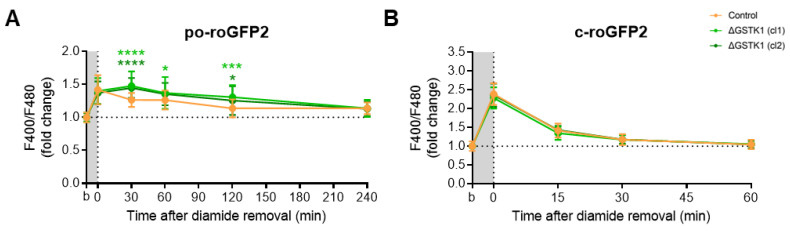
GSTK1 aids in the recovery of po- but not c-roGFP2 after diamide withdrawal. CT or ∆GSTK1 (cl 1 or 2) HEK-293 cells were transfected with a plasmid encoding (**A**) po-roGFP2 or (**B**) c-roGFP2 and cultured in rMEMα. After two to three days, the basal oxidation states (b) of po- and c-roGFP2 were measured and the cells were incubated (**A**) for 10 min with 2 mM diamide and 0.001% (*w*/*v*) Triton X-100 or (**B**) for 5 min with 50 µM diamide. After the removal of the oxidant, the F400/F480 response ratios of the sensors were monitored over time. The response ratios were normalized to the average value of the corresponding basal condition. The data points and vertical bars represent the mean and standard deviation of 30 to 60 individual measurements from 3 to 6 independent experiments, respectively. For each time point, the data obtained for the ∆GSTK1 conditions were statistically compared with those of the CT cells using the ordinary two-way ANOVA test with Tukey’s multiple comparisons test (*, *p* < 0.05; ***, *p* < 0.001; ****, *p* < 0.0001).

**Figure 7 antioxidants-12-01236-f007:**
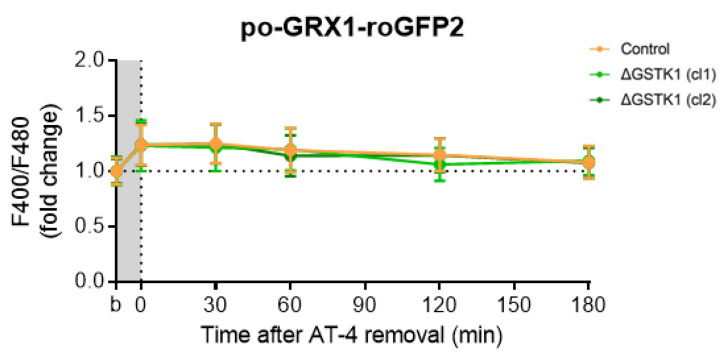
Fusion of GRX1 to the N-terminus of po-roGFP2 rescues the recovery phenotype of the glutathione redox sensor in AT-4-insulted ΔGSTK1 cells. CT or ∆GSTK1 (cl 1 or 2) HEK-293 cells were transfected with a plasmid encoding po-GRX1-roGFP2 and cultured in rMEMα. After two to three days, the basal oxidation states (b) of po-GRX1-roGFP2 were measured, and the cells were incubated for 10 min with 2 mM AT-4. After the removal of the oxidant, the response ratios of the sensors were monitored over time and normalized to the average value of the corresponding basal condition. The data points and vertical bars represent the mean and standard deviation of 30 individual measurements from 3 independent experiments, respectively. For each time point, the ∆GSTK1 data were statistically compared with those of the CT cells using the ordinary two-way ANOVA test with Tukey’s multiple comparisons test, but no significant differences were observed.

**Figure 8 antioxidants-12-01236-f008:**
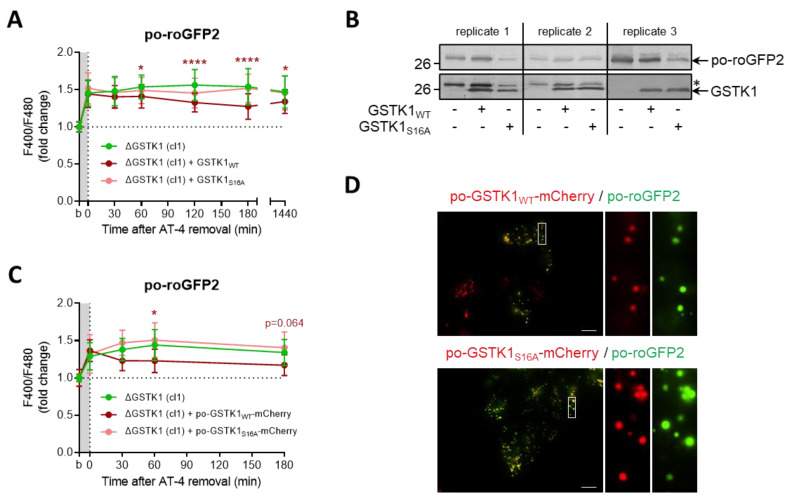
The GSTK1-mediated recovery of po-roGFP2 after exposure to AT-4 depends on the active-site serine residue. ∆GSTK1 (cl 1) HEK-293 cells were (co-)transfected with a plasmid encoding po-roGFP2, in combination or not with a plasmid encoding (**A**,**B**) GSTK1_WT_ or GSTK1_S16A_, or (**C**,**D**) po-GSTK1_WT_-mCherry or po-GSTK1_S16A_-mCherry, and cultured in rMEMα. (**A**,**C**) After two to three days, the basal oxidation states (b) of po-roGFP2 were measured, and the cells were incubated for 10 min with (**A**) 1 mM AT-4 or (**C**) 2 mM AT-4. After the removal of the oxidant, the response ratios of the sensors were monitored over time and normalized to the average value of the corresponding basal condition. The data points and vertical bars represent the mean and standard deviation of (**A**) 30 or (**C**) 10 individual measurements from three or one independent experiments, respectively (for panel **C**, only mCherry- and roGFP2-positive regions of interest were selected). For each time point, the results of the complementation analyses were statistically compared with those of the noncomplemented cells using the ordinary two-way ANOVA test with Tukey’s multiple comparisons test (*, *p* < 0.05; ****, *p* < 0.0001). (**B**) Total cell lysates from cells similar to the ones used in panel A were processed for immunoblot analysis with antisera directed against GSTK1 or EGFP (as the blots of replicate 1 and replicate 2 were first probed with the anti-EGFP antibody, the protein bands indicated by the asterisk represent po-roGFP2). The migration points of relevant molecular mass markers (expressed in kDa) are shown on the left. (**D**) After two days, cells were processed for live-cell imaging (scale bar: 10 µm).

**Figure 9 antioxidants-12-01236-f009:**
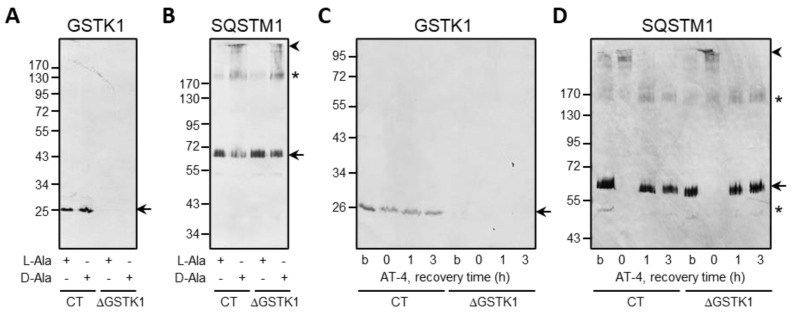
Po-H_2_O_2_ and AT-4 do not induce protein disulfide bond formation in GSTK1. CT or ΔGSTK1 (cl 1) HEK-293 cells were (**A**,**B**) cultured for three days in rMEMα containing 1 µg/mL doxycycline and 500 nM Shield1 (to express and stabilize DD-DAO, respectively), chased for one day in the same medium lacking doxycycline and Shield1 (to degrade the residual cytosolic pool of DD-DAO that has not yet been imported into peroxisomes), and incubated for 1 h in DPBS containing 10 mM 3-AT and 10 mM L-Ala or D-Ala, or (**C**,**D**) incubated for 10 min in rMEMα supplemented or not (b, basal state) with 500 µM AT-4 and chased in the same medium without AT-4 for 0, 1, and 3 h. Next, the cells were treated with 10 mM NEM (to block free thiol groups), and total cell lysates were processed for SDS-PAGE under nonreducing conditions and immunoblot analysis with antibodies specific for GSTK1 and SQSTM1. The migration points of relevant molecular mass markers (expressed in kDa) are shown on the left. The arrows and arrowheads mark nonoxidatively and oxidatively modified immunoreactive protein bands, respectively. The asterisks mark bands of unknown nature.

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
