# Peer review of "Functional Analysis of GSTK1 in Peroxisomal Redox Homeostasis in HEK-293 Cells"

_antioxidants, 2023, doi:10.3390/antiox12061236_

Round 1

Reviewer 1 Report

The paper of Costa et al. investigated the role of Glutathione-S-Transferase (GSTK1) enzyme in peroxisomal glutathione regulation. Interestingly, one of the hallmarks of peroxisomes  is their capacity to produce by their numerous oxidases’ hydrogen peroxide, which is degraded by catalase. However, except catalase admitted as a major antioxidant peroxisomal enzyme, little is known about how peroxisomes manage their redox balance. Here, authors, using an elegant GSTK1-deficient HEK-293 cell line model, they showed that the absence of peroxisomal GSTK1 expression has no effect on the basal redox state in peroxisome compartment but seams to be involved in the delayed response of peroxisome to thiol-specific oxidants.

The introduction is well written and sufficient to understand the rational of the study.

Materials and methods section describe all methods used, including materials suppliers.

Results are presented in a logical progression and sustained the issued arguments that are discussed under the light of the published data.

In this study, among several interesting conclusions, one underline a GRX-like activity in peroxisome. This was evidenced by the experiment (Figure 6) showing the recovery phenotype of the glutathione redox sensor in AT-4-insulted ΔGSTK1 cells by a the chimeric GRX1-po-roGFP2 protein. As the authors obtained a purified peroxisomal and mitochondrial fractions by fractionation, one could aske if the authors have measured a GRX and GSTK1 activities in these fractions. This would be of great interest to interrogate the presence of such activities in peroxisome and their distribution between peroxisomal and mitochondrial compartments.   

Reviewer 2 Report

1. The meaning of the abbreviations PO, MT, ER in the figure should be included in the figure notes or descriptions of Figure 1B.

2. The legend of Figure 2 A and B is not clear, A means that the knockout does not affect the expression of peroxisome-related, B means that it does not affect the expression of mitochondrial proteins, but the figure notes and the analysis mix A and B, the description is not clear enough.

3. Figure 2 A and B, the paper explains that cl 1 and cl 2 have no influence on the expression of ACAA 1 and CAT, but it is clear from the figure that there is a difference in ACAA1 and CAT protein expression between cl1 and cl2. Is this difference related to the fact that cl2 is homozygous?

4. In Figure 4, "Importantly, reintroducing GSTK1 into the ∆GSTK1 cells restored this deficit (Figure 4B,D) ". But there seems to be no difference in the expression of po-roGFP2 between ∆GSTK1 and ∆GSTK1+GSTK1 groups in 4D, but there is a significant difference between these two groups in 4B, can the inconsistent results prove the same conclusion?

5. The phrase "SDS-PAGE, Western blotting and immunoblot analysis" appears several times in the paper and refers to the same experimental technique. There is no need to repeat it three times.

6. The problem of long figure legends is common in figures, which can be refined, e.g. some descriptions related to methods should be omitted. In Fig. 6 and Fig. 7, try to avoid citing Fig. 4 in the figure notes; while the figure notes are too long, the meaning of the expression is not clear and it is recommended to revise them.

7. Throughout the article, most of the cl1 and cl2 groups are involved, but in Fig. 2 there is no analysis of the difference between cl1 and cl2 in the study of WB, and there is almost no difference between cl1 and cl2 in the experimental results Fig. 4 - Fig. 6. Fig4 A and B are essentially the same experiment, but there is no cl2 in B. Can the cl2 group be considered optional? The study of basal oxidation states (b) does not reflect the importance of designing cl1 and cl2 as heterozygous and homozygous controls, and there is no cl1 and cl2 in Fig 8. It is possible to briefly describe the significance of whether the two groups are set and what they can demonstrate.

8. There are some abbreviated terms in the text which appear for the first time without explanation.

9. Is it thin evidence for rejecting the previous view by judging that GSTK1 is not present in the endoplasmic reticulum based only on Fig 1 B, especially a band of GSTK1 appears in the endoplasmic reticulum region in the figure? Perhaps speculate on the reason for the presence of GSTK1 in the endoplasmic reticulum? Or could you consider locating it from different angles? Or try to separate the endoplasmic reticulum from the mitochondria completely?

10. The text in Fig. 8 it is simply stated that SQSTM1 is used as a positive control without explaining the reason. A sentence should be added like that disulfide bonds may mediate SQSTM1 multimerisation and therefore be used as a positive control to investigate whether oxidative damage causes GSTK1 to form disulfide bonds. Otherwise, there may be more than disulfide bonds that can cause SQSTM1 multimerisation and the figure only shows that oxidative damage does not cause GSTK1 multimerisation and does not reflect the role of disulfide bonds.

Minor editing of English language required
